# Effect of Liquefaction of Honey on the Content of Phenolic Compounds

**DOI:** 10.3390/molecules28020714

**Published:** 2023-01-11

**Authors:** Tomáš Hájek

**Affiliations:** Department of Analytical Chemistry, Faculty of Chemical Technology, University of Pardubice, Studentská 95, 532 10 Pardubice, Czech Republic; tomas.hajek@upce.cz

**Keywords:** phenolic compounds, flavonoids, antioxidants, honey, liquid chromatography, coulometric detection, microwave

## Abstract

Thermal liquefaction at low temperature is very time consuming and microwaves or an ultrasonic bath can be used to accelerate the process of dissolving sugar crystals. Phenolic compounds, such as phenolic acids or flavonoids, are an important group of secondary metabolites of plants and become honey from the nectar of blossoms. In this study, how the content of phenolic acids and flavones in honey were affected by liquefaction of honey using a microwave oven was studied. The concentration of tested compounds in untreated honey and in honey liquefied in a hot water bath, ultrasonic bath and microwave oven at four microwave power levels were determined by reversed phase liquid chromatography combined with multichannel electrochemical detection. A significant decrease in the content of all compounds was observed for all melting treatments. The phenolic compounds concentration decreased on average by 31.1–35.5% using microwave at intensities 270, 450 and 900 W and the time required for the sugar crystal melting was more than 20 times less than in the case of the 80 °C water bath. The temperature of samples after the end of microwave liquefaction was 76–89 °C. Significantly higher losses of phenolic compounds were observed during ultrasound treatment (48.5%), although the maximum temperature of honey was 45 °C, and at the lowest microwaves power (50.6%).

## 1. Introduction

Honey is a sweet and viscous natural foodstuff produced by honeybees or other insects from sugary secretions of plants (nectar) through regurgitation, enzymatic activity and water evaporation (honeydew) [1]. Honey is often used as sugar substitute, an ingredient or natural preservative in foods [2] and is well known for its high nutritional prophylactic medicinal value [3]. Sugars represent the larger part of honey, typically 70–80% (*w*/*w*). The major carbohydrates are the monosaccharides glucose (28–41%, *w*/*w*) and fructose (22–35%, *w*/*w*) [4]. Except for the sugars, honey is composed of other compounds such as enzymes, amino acids, organic acids, vitamins soluble in water, minerals and aromatic compounds. Honey is rich in phenolic acids and flavonoids which act as natural antioxidants, eliminating free radicals and inhibiting cell membrane lipids oxidation and protection of deoxyribonucleic acid (DNA), having antimicrobial, or anticancer activity [5,6]. The phenolic compounds in honey originate from the plants that are used to collect the nectar by honeybees, so the content of phenolic compounds depends on the botanical and geographical origin of honey [7].

In terms of structure, phenolic acids can be divided into two subgroups according to their structure: the hydroxybenzoic and hydroxycinnamic acids derivatives, and they are important compounds with bioactive functions occurring in vegetable products and foods. Numerous phenolic acids were identified in honey samples, for example caffeic acid, gallic acid, cinnamic acid, *p*-coumatic acid, protocatechiuc acid, 4-hydroxybenzoic acid, 4-hydroxyphenylacetic acid, vanillic acid, syringic acid, ferulic acid and chlorogenic acid [8,9]. Flavonoids represent the largest group of plant polyphenolic compounds and are widely distributed in leaves, flowers, trees and seeds. Flavonoids can significantly contribute to the overall antioxidant capacity of honey and bringing beneficial effect for human health [10]. The total flavonoids content in honey represented 2–10% (*w*/*w*) of the total phenolic content [11]. For example, naringenin, hesperetin, pinocembrin, chrysin, galangin, lutheolin, rutin, quercetin and keampferol were found in honey samples [9,12]. The composition of compounds, color, aroma and flavor of honey depend mainly on the flowers, geographical regions, climate and honeybee species involved in its production [13], but specific flavonoids have been described as markers of the botanical origin of unifloral honeys [14]. The disadvantage of the phenolic compound is their poor stability and the rate of degradation in food during heat processing or during storage depends not only on temperature but also on pH and the presence of oxygen or light [15,16]. 

Due to the physical properties of the phenolic acids and flavonoids, liquid chromatography in reverse phase mode (RP-LC) is one of the more effective methods for separation and determination these individual compounds. The octadecyl (C18) stationary phases provide excellent separation selectivity [17,18,19] but also other functional ligands bind to silica separate phenolic acids and flavonoids sufficiently: pentafluorophenyl (PFP) [20] or amide C16 [3]. After separation, an ultraviolet (UV) detector [17,21,22], including a diode array detector (DAD), mass spectrometry (MS) [9,17,20] or electrochemical detection (ECD) [18,19,23,24] can be used for monitoring compounds leaving the column. ECD offers better sensitivity and high selectivity compared to UV detection. The CoulArray is coulometric detector that simultaneously records the current responses of eight flow-through cells connected in series, with different applied potentials. The ratios of the areas of the pre-dominant and post-dominant peaks to the area of the dominant peak (recorded in the channel providing the highest response) can be used to improve identification of compounds [25]. The preparation of the honey sample for separation consists of diluting it with acidified water and concentrating the phenolic acids and flavonoids on nonpolar [10,26,27] or ion-exchange [28] solid phase extraction (SPE) cartridge. The SPE step is used not only to concentrate analytes, but also to remove sugars and other highly polar compounds [21].

Crystallization of honey is a natural process, due to supersaturated solution of glucose, and every honey passes through the crystallization sooner or later. The rate of crystallization, commonly called granulation, is affected by many factors, but the primary factor is the ratio of fructose to glucose. Honeys with very high percentage of glucose crystallize almost immediately after harvesting [29,30]. On the other hand, higher concentration of fructose delays the crystallization. In general, the nectar honeys crystallize earlier than honeydew honeys and the rate of crystallization is influenced by the presence of crystals, pollen and dust grains.

The crystallized honey can be relatively easily melted without losing quality. The melting point of crystallized honey is between 40 and 50 °C, depending on sugar composition. The time required to melting of honey at 50 °C is very long, for several hours or even tens of hours, and higher temperatures usually damages the thermolabile components of honey, such enzymes or phenolic compounds. Furthermore, undesirable reactions may occur (e.g., browning and changes in colour, off-flavours), food contaminants are produced (e.g., 5-hydroxymethyl-2-furfural) and the honey loses quality [31,32]. 

The application of ultrasound has been shown to eliminate existing crystals and also retard the crystallisation process and completely destroy yeast, mold and coliform bacteria [33]. Consumers at home usually do not have time to melt the honey at low temperature for many hours and do not have an ultrasound bath available, but use higher temperature or a microwave oven. The aim of the present study was to investigate, whether melting of honey at a higher temperature than 50 °C, melting in a microwave oven or in an ultrasonic bath affects the content of phenolic acids and flavonoids.

## 2. Results and Discussion

### 2.1. Development of HPLC-CoulArray Method

The HPLC-CoulArray method was developed for simultaneous identification and quantification of 13 phenolic acids, 4 phenolic aldehydes and 4 flavonoids (Figure 1) occurring in a honey. A core-shell C18 column and acidified water/acetonitrile mobile phase under gradient elution was performed to separate all tested compounds in 40 min (Figure 2). The retention times are in Table 1. The pH 3.1 of the mobile phase provided the best resolution and peak shape for separation phenolic acids due to suppression of dissociation [25]. Eight different potentials (200–900 mV) were set on the flow cells of coulometric CoulArray detector and the channel providing the highest response was used for quantification (Table 1). Twelve tested compounds reached maximum response at 900 mV. Flavonoids myricetine and quercetin showed the response only at a low potential (300 mV) while rutin, the glycoside of quercetin, was oxidized at 400 mV and above.

The CoulArray coulometric detector is very sensitive with a low detection limit and most samples can be analysed without prior treatment. However, the honey samples contain a large amount of sugars that can negatively affect separation and the stationary phase in the column. SPE Strata C18-E column was an effective tool for removing sugars and all high polar compounds and at the same time the analytes were concentrated. 

The sorption capacity of the SPE column for analytes was not exceeded, which was verified by analysis of last 2 mL of diluted sample loaded on the SPE column. Equally, additional 1 mL of methanol was applied, collected and analyzed after the SPE step to verify that a sufficient volume of eluent (2 mL of methanol) was used. Problems with quantitation in SPE may be associated with an inability to recover target analytes. For this reason, we tested recovery for all tested compounds including the matrix effect on a sample of honey spiked with standard mixture. Recovery SPE column was 70–100% for most tested compound (Table 1) except gentisic acid, β-resorcylic acid and salicylic acid, however these compounds have not been identified in honey anyway. The relative standard deviations of recovery (*n* = 4) did not exceed 5% (Table 1).

The sensitivity of the method was proved with the limit of detection (LOD) and the limit of quantification (LOQ) (Table 1). The LOD ranged from 1 to 16.3 µg/kg and the LOQ ranged from 6.4 to 54.5 µg/kg of honey.

### 2.2. Temperature and Duration of Individual Types of Honey Treatment

Three types of liquefaction of crystallized honey were used to dissolve sugar crystals in honey: water bath, ultrasonic bath and microwave oven. Two different temperature range of water bath (60 °C and 80 °C) and four microwave wattage range (90, 270, 450 and 900 W) were tested. During the liquefaction process, the honey was stirred every 30 s in case of microwave treatment, and every 5 min in case of water or ultrasonic bath. Stirring contributed to the optimal distribution of the heat distribution and thus to the faster melting of the crystals. Hot centers are formed especially in the case of the use of microwaves. The honey sample was melted until all the crystals were completely dissolved. The time required to dissolve all the crystals varied, depending on the type of treatment (Table 2). In addition, the temperature of the honey sample was measured with a non-contact thermometer (Table 2). 

The temperature of the sample in the ultrasonic bath did not exceed 45 °C and the time required for melting was the longest compared to the other methods (90 min). The final temperature of honey in glass bottle after microwave liquefaction was comparable or higher than the temperature of the water bath at 80 °C and was between 76 and 89 °C. The lowest intensity of microwaves (90 W) had to be employed disproportionately longer (11 min) for disappearance of all sugar crystals than in the case of higher powers of the microwave oven (0.33–1 min). The temperature at 90 W increased slowly until the final temperature of 79 °C was reached. According to the sound of the microwave oven at 90 W, the microwaves were emitted for 3 s every 29 s (the sample was not irradiated for 26 s). Thus, the sample was exposed to microwave radiation each cycle between stirring which would also correspond to an increase in temperature (Appendix A). The short time at higher microwave intensities may be attributed to the formation of centers with a higher temperature than was measured after liquefaction. Moreover, the highest radiation intensity (900 W) caused a very rapid heating of the sample after only 30 s, therefore the sample was mixed every 10 s. In the case of low intensity, these centers do not occur, and the heat is better dispersed throughout the volume. The exposure time at 60 °C was more than one hour (65 min) and the increased temperature to 80 °C accelerated the dissolution three times (20 min).

### 2.3. The Effect of Type of Liquefaction on Antioxidant Content

All liquefied honey samples as well as the untreated honey sample were analyzed by the HPLC-CoulArray method for presence of phenolic acid, phenolic aldehyde and flavonoid compounds. Six of the tested compounds (protocatechuic acid, protocatechuic aldehyde, 4-hydroxyphenylacetic acid, chlorogenic acid, rutin and ethylvanillin) were identified in samples of honey on the basis of position in the chromatogram, ratios of oxidation current over a range of working cells potential and based on standard addition. The quantity of these compounds was subsequently monitored in liquefied honey. 

The content of identified analytes in untreated honey, honey melted in (i) hot water bath, (ii) ultrasonic bath, and (iii) microwave oven is presented visually in Figure 3. Protocatechuic acid and chlorogenic acid was found in the highest quantity (1461.1 µg/kg respective 832.4 µg/kg). The concentration level of 4-hydroxyphenylacetic acid and rutin was similar (215.9 µg/kg and 204.3 µg/kg) and protocatechic aldehyde and ethylvanillin was found in small amount (137.4 µg/kg respective 90.8 µg/kg). The presence of ethylvanillin may indicate the flavoring of honey, because this compound does not occur naturally in nature. For this reason, ethylvanilin was added to the tested mixture.

The effect of liquefaction of honey on the content of the monitored phenolic acids and flavonoids was clearly visible and result was a decrease in the concentration of antioxidants present in honey (Figure 3). The determined concentration together with standard deviations are given numerically in the Appendix A. Statistical evaluation of differences between honey without treatment and individual methods of liquefaction was performed by multiple pairwise comparisons using Duncan’s test at the statistical level of significance *p* < 0.05. 

The content of protocatechuic acid decreased the most for all liquefaction methods, by approximately 60–70% (Table 3). The water bath at 60 °C and 80 °C and a microwave oven with at 270 W and 450 W caused statistically the same loss of protocatechuic acid. The smallest concentration of protocatechuic acid, and statistically significantly different from the others, was determined in honey liquefied in an ultrasonic bath and in a microwave oven at 900 W. The greatest differences in loss were observed for chlorogenic acid (24–62 %). For chlorogenic acid, melting in a water bath was the gentlest method of liquefaction, while ultrasonic bath and microwave were the most extreme method of liquefaction, resulting in the highest concentration decrease. The concentration of rutin, as a representative of the glycoside of flavonoids, was significantly lower in melted honey than in the original untreated honey (decrease 36–69%). A statistically significant difference from the other methods was observed for the microwave method with low wattage (90 W) when the content of rutin decreased by up to 70%. Ethylvanillin showed similar results as rutin. The significance of differences was not detected with the use of a water bath and microwave oven (loss 25–30%), except microwaves at the lowest intensity (loss 51%).

On the other hand, the concentration of procatechuic aldehyde and 4-hydroxyphenylacetic acid were not significantly affected by the use of a water bath at 60 °C and microwave oven at 270–900 W. Moreover, the amount was slightly increased (5.5% and 11.7% respectively) when using a water bath at 40 °C, but statistically insignificant compared to honey without heating. However, water bath at 80 °C, ultrasonic bath and microwave again reduced content by 44–53% in case of procatechuic aldehyde and 16–46% in case of 4-hydroxyphenylacetic acid. 

In summary, liquefaction of honey into liquid form had a negative impact on the content of all phenolics and flavonoids and the gentlest method we tested was heating in a water bath at 60 °C despite the fact that the honey was exposed to this temperature for a long time (65 min). During melting, the concentration of compounds decreased by an average of 25.3% (Table 3). Increasing the water bath temperature by 20 °C resulted in a further reduction of approximately 15 %. Results are consistent with the conclusions of Villacrés-Granda [34], which determined total phenolic content, total flavonoid content or antioxidant activity for honeys heated to 45 °C and 60 °C. The total phenolic content decreased by 44% and antioxidant capacity (determined by Ferric Reducing Antioxidant Power assay) decreased by 31% in honeys treatment at 60 °C. 

Liquefaction in microwave oven at 270, 450 and 900 W affected the content of tested compounds identically (reduction of 31.1–35.5%) and the wattage used in this process is not considered. After this treatment, the sample temperature was 76–89 °C and thus comparable to the temperature of 80 °C water bath. The exposure time in the water bath was significantly higher than in the microwave oven (more than 20 times) which was also reflected in a further decrease in the content of tested compounds (39.9%). 

The treatment with ultrasonic bath and microwave oven at 90 W showed the least sensitivity to the sum of phenolic acids and flavonoids. Almost 50% was lost. The reason for the considerable decrease in the concentration of compounds may be explained in the time of treatment. The crystal dissolution of honey in the ultrasonic bath was the disproportionately long time (90 min) in comparison with other methods and is the likely reason for such a large loss of compounds even though the sample temperature was not higher than 45 °C. Similarly, the duration of melting honey with lower wattage microwaves required a longer time than microwaves with higher wattage (11 min and less than 1 min respectively). In this case, the time is certainly a decisive factor affecting the content of phenolic compounds because the final temperature (79 °C) was comparable to other microwave powers (76–89 °C). It can be noted that, when using a microwave oven for liquefaction of crystallized honey, the power setting is not decisive for the content of phenolic acids and flavonoids. 

In conclusion, the content of phenolic compound should not be the only criterion for selecting the method of liquefaction of honey, but also other properties of honey, for example antibacterial activity, enzymatic activity, sensory properties of honey or hydroxymethylfurfural (HMF) content (the main indicator of overheating during processing), should be considered. According to available publications, the use of microwave radiation to liquefy honey results in a significant reduction in antibacterial activity [33] and inactivation of diastase enzymes is much faster under microwave treatment than in the conventional process [34]. On the other hand, shorter duration at higher power intensity was desirable in terms of lower HMF value and higher diastase activity [35,36].

## 3. Materials and Methods

### 3.1. Chemical and Standards

Acetonitrile (gradient grade), methanol (HPLC grade), ammonium acetate, formic acid (≥99.8%) and all phenolic acids and flavonoids (Figure 1) were purchased from Sigma-Aldrich (St. Louis, MI, USA). Deionised water was treated with a Milli-Q Reference Water Purification System (Millipore SAS, Molsheim, France). 

### 3.2. Equipment

A HPLC system for phenolic compounds analysis consisted of a vacuum degasser DG 3014 (Ecom, Prague, Czech Republic), two chromatographic pumps model 582, a CoulArray thermostatic organizer (both ESA, Chelmsford, MA, USA) containing a pulse damper, a gradient mixer, a manual injector with a 10 mL sampling loop (Rheodyne, Cottati, CA, USA); an electrochemical 8-channel CoulArray 5600 A detector and a PC with a CoulArray software for data acquisition, processing and analysis (both ESA Chelmsford, MA, USA). A chromatographic core-shell column Ascentis Express C18 (150 mm × 3 mm I.D., 2.7 µm particle size) was obtained from Phenomenex (Torrance, CA, USA). The temperature of the column and the detector array was maintained at 40 °C. The mobile phase consisted of 1% acetonitrile in 10 mM ammonium acetate adjusted to the pH 3.1 by adding a few drops of formic acid (A) and pure acetonitrile (B). Before use, the mobile phases were filtered over a Millipore (Bedford, MA, USA) 0.45 µm filter. The flow rate of mobile phase was set to 0.4 mL/min and the gradient elution was used as follows: 0 min—0% B, 10 min—0% B, 20 min—14% B, 25 min—19% B, 40 min—34% B. Working potentials of 250, 300, 400, 500, 600, 700, 800, and 900 mV were applied at the eight electrochemical cells of the CoulArray detector.

### 3.3. Methods

Flower honey sample (blend of EU and non-EU honeys, producer JSG med corp., Pilsen, Czech Republic) was obtained from the trade network. Three methods of liquefaction of honey were tested: a warm water bath, an ultrasonic bath and a microwave oven. A total of 30 g of crystallized of honey was sealed in a 15 mL clear glass bottle. The sample of honey was placed in a 60 °C or 80 °C water bath Julabo EC-5 (Julabo GmbH, Seelbach, Germany), in an ultrasonic bath Bandelin sonorex RK 52 H (Banderin electronic, Berlin, Germany) or a microwave oven (Gallet FMOEG 251 W, Gallet, Fagnières, France) until complete liquefaction of honey. The ultrasonic rated power was 60 W at frequency 35 kHz. The maximum rated power of the microwave oven was 900 W and the microwave frequency was 2450 Hz. Four microwave intensities were tested: 90, 270, 450, 900 W. During liquefaction in microwave oven, the honey was stirred every 30 s by vigorous shaking (approximately 10 s) to disperse the temperature and sugar crystals. In case of ultrasonic bath and water bath, samples were stirred every 5 min. The temperature of samples was measured by using a non-contact infrared thermometer Testo 830-T1 (Testo, Titisee-Neustadt, Germany).

For all samples (honey before and after liquefaction), 5 g of honey was mixed with 20 mL of deionized water adjusted to pH 3 with 1M HCl. SPE extraction was performed with Visiprep vacuum manifold (Sigma-Aldrich (St. Louis, MI, USA). The cartridge Strata C18-E (55 µm, 70 A, 500 mg) purchased from Phenomenex (Torrance, CA, USA) was conditioned with 5 mL of methanol and 5 mL of deionized water. The diluted sample was passed through the SPE cartridge, then washed with 5 mL of deionized water to remove all sugars and eluted with 2 mL of methanol. The collected 2 mL of concentrated sample was diluted with 1 mL of water. The extracts were filtered through the 0.45 mm PTFE syringe filter and injected to HPLC system. The retention times and the behaviour of compounds recorded at different applied potentials were used for identification of compounds. Compounds were quantified in using external standard calibration method. The calibration standard solutions were prepared in the concentration range from 0.01 µg/mL to 10 µg/mL and the regression parameters are listed in the Appendix A.

## 4. Conclusions

Conversion of crystallized honey back to liquid form can be performed in several ways. The recommended heating to a temperature of 45–50 °C is time consuming and not very attractive for home use. The present study demonstrated that thermal liquefaction at higher temperature (80 °C) reduces the content of phenolic acid and flavonoids in the same way as microwave energy, approximately 30–40% of the tested antioxidants were lost on average. However, microwave heating dissolved the sugar crystals significantly faster than a water bath even though the honey was heated to a similar temperature. The highest intensity of the microwave heats the honey too quickly and the treatment process is difficult to control. Conversely, very low power prolongs the effect of microwave radiation on honey and the content of phenolic compounds is significantly reduced. From the point of view of the content of phenolic compounds, the most suitable treatment is microwave method at medium intensity. The negative effect of microwave radiation on other components of honey should be considered before deciding to use a microwave oven to liquefy honey.

## Figures and Tables

**Figure 1 molecules-28-00714-f001:**
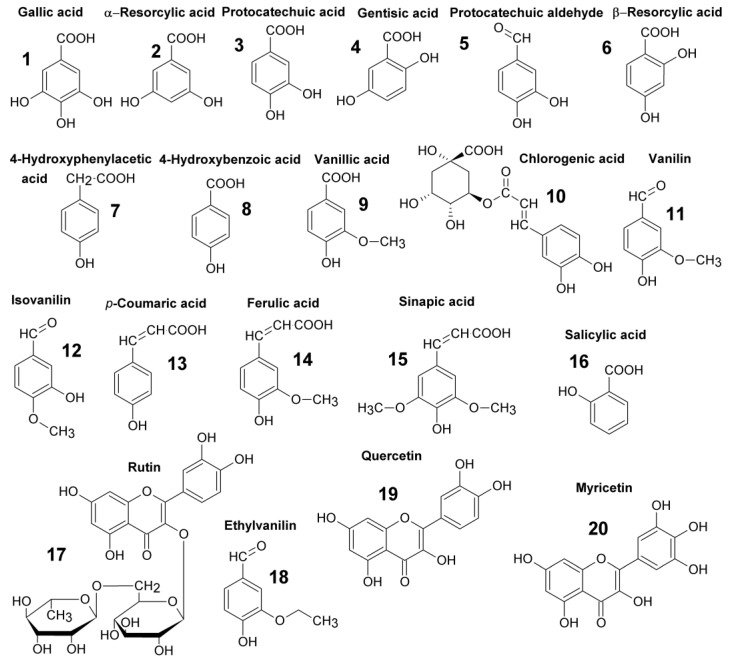
Structures of the phenolic acids, phenolic aldehydes and flavonoids.

**Figure 2 molecules-28-00714-f002:**
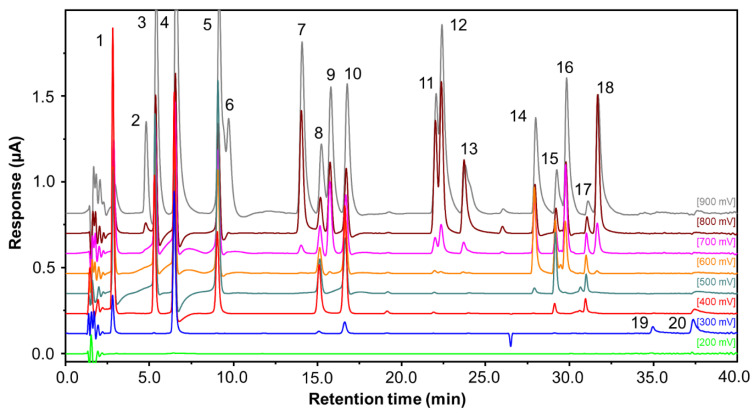
RP-LC-CoulArray chromatogram of phenolic acid and flavonoid standards. Peak numbers correspond to the numbers presented in Table 1.

**Figure 3 molecules-28-00714-f003:**
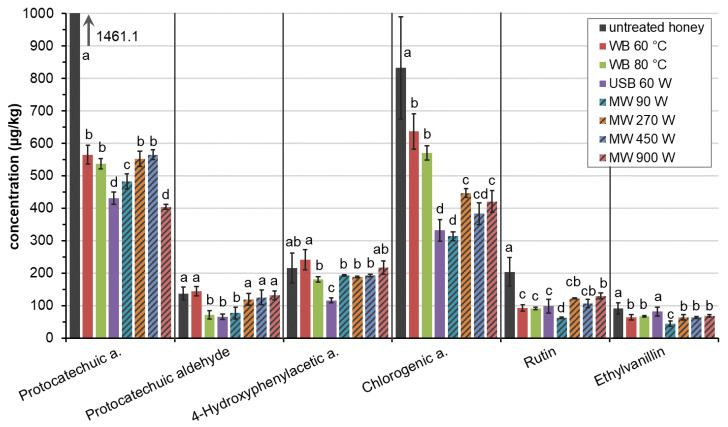
The effect of type of liquefaction of honey in hot water bath, ultrasonic bath, and microwave oven o on antioxidant content. MW—microwave, Error bars—standard deviation (*n* = 4), a,b,c,d—statistically significant differences of Duncan’s test (*p* < 0.05). Different letters mean statistically significant differences, and conversely the same letter means an insignificant difference between the values. Values within a given group (type of treatment), which do not differ statistically significantly are marked with the same letters.

**Table 1 molecules-28-00714-t001:** Retention time (*t_r_*), potential of the dominant peak (PDP), SPE recovery with RSD (*n* = 4), limit of detection (LOD) and limit of quantification (LOQ) of tested phenolic acids, phenolic aldehydes and flavonoids.

No.	Compounds	*t_r_* (min)	PDP (mV)	SPE Recovery (%)	SPE RSD (%)	LOD (µg/kg)	LOQ (µg/kg)
1	Gallic acid	2.72	400	79.42	1.84	15.1	50.1
2	α-Resorcylic acid	4.82	900	83.94	1.40	5.7	12.3
3	Protocatechic acid	5.34	900	71.98	4.74	14.3	54.8
4	Gentisic acid	6.44	900	62.33	3.65	10.0	25.4
5	Protocatechuic aldehyde	9.03	500	99.86	4.47	16.0	53.4
6	β-Resorcylic acid	9.49	900	68.24	2.63	5.0	11.6
7	4-Hydroxyphenylacetic acid	14.06	900	95.64	2.73	6.8	22.9
8	4-Hydroxybenzoic acid	15.11	900	91.34	2.17	8.9	30.8
9	Vanillic acid	15.85	900	86.77	1.42	6.2	23.3
10	Chlorogenic acid	16.81	900	95.78	3.26	4.3	14.4
11	Vanillin	22.03	900	88.20	1.14	16.3	45.1
12	Isovanillin	22.48	900	83.17	3.13	7.6	18.7
13	*p*-Coumaric acid	23.71	900	85.76	2.38	1.0	6.4
14	Ferulic acid	27.97	600	93.39	4.00	8.3	25.6
15	Sinapic acid	29.31	500	70.92	3.99	10.3	32.5
16	Salicylic acid	29.87	900	56.40	3.01	9.8	22.6
17	Rutin	31.12	500	90.67	4.78	2.9	9.8
18	Ethylvanilline	31.66	800	82.52	3.54	10.0	33.2
19	Myricetin	34.92	300	88.34	3.79	1.6	7.4
20	Quercetin	37.41	300	85.46	3.78	15.1	50.1

**Table 2 molecules-28-00714-t002:** Temperature and duration of individual types of honey treatment. WB—water bath, USB—ultrasonic bath, MW—microwave (temperature after all crystal melting).

Parameter	WB 60 °C	WB 80 °C	USB 60 W	MW 90 W	MW 270 W	MW 450 W	MW 900 W
time (min)	65	20	90	11	1	0.50	0.33
temperature (°C)	60	80	45	79	82	76	89

**Table 3 molecules-28-00714-t003:** Percentage changes in concentration of tested compound compared to honey without treatment and average change in concentration. WB—water bath, USB—ultrasonic bath, MW—microwave.

Compound	WB 60 °C (%)	WB 80 °C (%)	USB 60 W (%)	MW 90 W (%)	MW 270 W (%)	MW 450 W (%)	MW 900 W (%)
Protocatechuic a.	−61.3	−63.2	−70.5	−67.0	−62.2	−61.4	−72.4
Protocatechuic aldehyde	5.5	−47.5	−52.5	−43.9	−13.3	−9.0	−3.9
4-Hydroxyphenylacetic a.	11.7	−16.2	−46.3	−10.3	−12.7	−10.8	0.6
Chlorogenic a.	−23.5	−31.4	−60.1	−62.3	−46.4	−53.9	−49.5
Rutin	−54.7	−55.2	−51.8	−69.5	−40.2	−47.6	−36.4
Ethylvanillin	−29.3	−25.9	−10.1	−50.8	−29.9	−30.4	−25.0
average changes	−25.3	−39.9	−48.5	−50.6	−34.1	−35.5	−31.1

## Data Availability

The data presented in this study are available on request from the authors.

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
