# Peer review of "Effect of Liquefaction of Honey on the Content of Phenolic Compounds"

_molecules, 2023, doi:10.3390/molecules28020714_

Round 1

Reviewer 1 Report

Reviewing of Effect of liquefaction of honey on the content of phenolic compounds

Reference. molecules-2113369-peer-review-v1

Overall appreciation

This study was carried out to assess the thermal liquefaction of honey at 60 and 80°C, by using microwaves at 90, 270, 450, 900 W; or an ultrasonic bath on phenolic content.

The paper is well written and is in agreement with the scope of the Journal. Some precisions should be added to material and methods section and considered in results and discussion, abstract and conclusion.

It will be appropriate for publication after revisions.

Recommendation: major revisions

Title of the manuscript (MS) associated with supplementary table S1 is not the same with that of the whole manuscript:

 Effect of liquefaction of honey on the content of phenolic compounds (MS)

and

Study of the influence of melting of crystalized honey on content of phenolic compounds (title with table S1)

Abstract

-to be revised and reformulated according to discussion to be made considering (T, t) of honey sample submitted at different treatments

-Lines 9-10: reformulate this sentence,

‘ Thermal liquefaction at 60 and 80°C at low temperature after few hours and microwaves or an ultrasonic bath can be used to accelerate the process 10 of dissolving sugar crystals’

**>60 and 80°C are not considered to be low temperatures.

Material and methods

-

Lines 266-271. Precisions should be added to this section:

-There are many types of honey, please indicate the type of honey and geographical origin.

-during different liquefaction process, sample were covered or exposed to light, this is also very important if exposure time is long.

-for the different used conditions for liquefaction please indicate approximate required time for liquefaction:

*For warm water bath at 60 °C and 80 °C

**For ultrasonic bath,

**For each microwave power : 90, 270, 450, 900 W;

-

Besides:

- for ultrasonic bath corresponding power should be indicated, it’s known that with increasing US power from 80 to 400 Watt, the temperature of the sample should be increase from ambient temperature to 40-45°C,

*For each microwave power: 90, 270, 450, 900 W; corresponding temperature of the sample should be cited. It could be calculated or measured.

-Please indicate if honey was stirred manually? every 30 seconds.

è  These precisions are mandatory to discuss variations according to real temperature of the sample at different conditions and corresponding exposure time

Supplementary data: table 1: please use the same abbreviation US or USB for ultrasound, idem for MW and honey (indicated that is liquefaction at ambient T, or untreated ???):

honey WB 60 °C WB 80 °C USB MW 90 W MW 270 W MW 450 W MW 900 W

Results and discussion

-Lines 175-180, this paragraph should be added as footnote to the corresponding table and not in RD section

-Line 226; write the whole word, word approx.. 15% without abbreviation;

*line 231-241 this section should be written and discussed considering the (T, t) of samples reached for different liquefaction operations as mentioned above in MM section.

Conclusion

-reformulate the following sentence line 288, because MW treatment s not less energy consuming than heating at 45-50°C

‘The recommended heating to a temperature of 45–50 °C is time and energy-consuming and not very attractive for home use.’

I’m not sure with this statement line 294-295, in fact 900 watt is already high enough. It’s better to say that high energy and maybe high temperature with short time treatments are the most suitable. The factor time should be included and controlled in the experimental design

‘Then, if a microwave oven is used to melt honey, a higher intensity of microwaves is more desirable to reduce the heat exposure time.’

Reviewer 2 Report

The manuscript describes the effect of honey liquification at different conditions on the content of biologically active phenolic acids and a flavonoid glycoside. The paper is well presented, and the methods used are reasonably described. The results are of interest since this process is frequently applied from a lot of consumers. Only several very small corrections should be made in order to improve the quality of the manuscript presentation.

-          In Figure 1 the name of p-cumaric acid should be corrected to p-coumaric acid. The names of all compounds in the Figure and in all Tables should be unified – it is better all names to be with capital letter. For example, in Table 1 α-resorcylic acid to be α-Resorcylic acid, etc.

-          In the caption of Table 1 – ”…………………..tested phenolic acid and flavonoid compounds.” should be revised to “……tested phenolic acids, phenolic aldehydes and flavonoids.

-          Line 44 – cinnamic acid is not a phenolic acid. Please, revise.

Reviewer 3 Report

Submitted manuscript “Effect of liquefaction of honey on the content of phenolic compounds” focused on effect of different honey liquefaction methods such as water bath, microwave and ultrasound on phenolic content.

This study lacks of novelty but has some potential to contribute to current state of knowledge. Ms needs substantial revision and additional experiment needs to be carried out.

I have several important comments that need to be clarified/corrected.

1.  The methodological part must be updated and more information must be stated about the methods. For examples, ultrasound treatment is not properly described (time, power, temperature during/after treatment). Moreover, author did not specify what type of honey was used. Was it only one sample or more honey samples. Following sentence “The sample of mead was obtained from the trade network” is most likely mistake since author did not work with the mead but with honey. It is crucial to test several type of crystallised honey such as rapeseed, sunflower or other honey types which ae characterized with prompt crystallization.  It is difficult to make conclusion based only one samples of honey type.

2. My major concern, however, is missing the qualitative data about honey before and after honey treatment. Some qualitative indicators need to be showed such as HMF, diastase activity etc.

3. Although honey can effectively be crystallised by microwave, author completely omitted the facts that biological properties can significantly be changed. It has been shown that microwave is hardly controlled method of honey liquefaction and has detrimental effect on enzymatic activity of honey proteins and overall antibacterial activity. Consumers also buy honey as a functional food and this fact needs to take into account.  

After reading the manuscript, it seems that the efficacy of microwave treatment is comparable with thermal treatment in term of polyphenol content and thus, it could be used as a faster and effective method for de-crystalisation. However, based on above mentioned comments I do not agree that microwave is suitable method for honey liquefaction.

4. Author used thermal treatment only at two temperature and it would be great to see the results where most used temperature 40 and 50°C will be applied for honey liquefaction and compare the polyphenol content. Time period for honey liquefaction will be most likely longer in compare to 60 and 80°C.

Round 2

Reviewer 1 Report

Added sentences and paragraphs (in blue) should be deeply revised for english (some typewriting mistakes) and for giving precisions and correct values concerning, treatment time, reached temperture, % of phenol loss. Please see some points to consider in the pdf file. 

Author Response

Please see the attachment. The responses are in the notes.

Reviewer 3 Report

Author addressed some raised issues, however, I have still doubts about the methodology and conclusion.  Author used only one commercially available honey sample with unknown physico-chemical parameters. It is absolutely crucial to determine the basic legislative parameter if new/re-new method for liquefaction is introduced. These important points have not been properly addressed.

Author Response

Comments and Suggestions for Authors

Author addressed some raised issues, however, I have still doubts about the methodology and conclusion.  Author used only one commercially available honey sample with unknown physico-chemical parameters. It is absolutely crucial to determine the basic legislative parameter if new/re-new method for liquefaction is introduced. These important points have not been properly addressed.

Response: Please note that we did not introduced new method of liquefaction of honey, it was basic research. The aim of the work was to find out how the phenolic compounds are affected if the consumer decides to liquefy honey, for example, a microwave oven. I focused only on phenols and honey sample was commercially available in the trade network. Therefore, honey had to be in accordance with the legislative requirements. It would certainly be beneficial to analyze more honey samples or to determine more parameters, but our view was the content of phenolic compounds. Moreover, several works on the effect of honey liquefaction on HMF or diastase content have already been published. I am convinced that the conclusions obtained in this manuscript will also be valid for other honeys. In the next study we will certainly include other parameters in the research plan.